# Porcine Reproductive and Respiratory Syndrome Virus Engineered by Serine Substitution on the 44th Amino Acid of GP5 Resulted in a Potential Vaccine Candidate with the Ability to Produce High Levels of Neutralizing Antibody

**DOI:** 10.3390/vetsci10030191

**Published:** 2023-03-03

**Authors:** Jong-Chul Choi, Min-Sik Kim, Hwi-Yeon Choi, Yeong-Lim Kang, In-Yeong Choi, Sung-Won Jung, Ji-Yun Jeong, Min-Chul Kim, Andrew Y. Cho, Ji-Ho Lee, Dong-Hun Lee, Sang-Won Lee, Seung-Yong Park, Chang-Seon Song, In-Soo Choi, Joong-Bok Lee

**Affiliations:** 1Laboratory of Infectious Diseases, College of Veterinary Medicine, Konkuk University, 120 Neungdong-ro, Gwangjin-gu, Seoul 05029, Republic of Korea; 2Careside Co., Ltd., Woolim Lions Valley A-B210, #146-8, Sangdaewon-dong, Jungwon-gu, Seongnam 13209, Gyeonggi-do, Republic of Korea; 3Southeast Poultry Research Laboratory, Agricultural Research Service, U.S. Department of Agriculture, U.S. National Poultry Research Center, Athens, GA 30605, USA; 4KU Research Center for Zoonosis, 120 Neungdong-ro, Gwangjin-gu, Seoul 05029, Republic of Korea

**Keywords:** porcine reproductive and respiratory syndrome virus, vaccine, GP5, glycosylation, neutralizing antibody

## Abstract

**Simple Summary:**

The slow and weak formation of neutralizing antibodies against the PRRSV is related to the N-linked glycosylation site surrounding the major epitope in the ectodomain of GP5 protein. To develop vaccine candidates that induce high levels of neutralizing antibodies, various previous studies have attempted to substitute the various glycosylation sites with other amino acids. In this study, we created a new vaccine candidate, vCSL1-GP5-N44S, by replacing the 44th asparagine amino with serine. In an animal trial, piglets inoculated with vCSL1-GP5-N44S showed no adverse effects and had a protective efficacy, including a high level of neutralizing antibodies after the challenge. Therefore, it has been demonstrated that substituting the 44th N-linked glycosylation site of the GP5 protein with serine is preferable among the various substitution. Additionally, vCSL1-GP5-N44S has shown its potential as a vaccine candidate.

**Abstract:**

N-linked glycans covering GP5 neutralizing epitopes of porcine reproductive and respiratory syndrome virus (PRRSV) have been proposed to act as a sheath blocking the production of neutralizing antibodies. Herein, we genetically engineered PRRSV with serine (S) substitution on the 44th asparagine (N) on the GP5 ectodomain of PRRSV-2 lineage-1. To evaluate the recombinant PRRSV, in vivo experiments were performed in piglets. The recombinant virus group showed no viremia until 42 days post-inoculation (dpi), and the rectal temperature and average daily weight gain were in the normal range at the same time point as the negative control group. On the 42 dpi, both groups were challenged with the wild-type virus. The recombinant PRRSV group showed lower rectal temperature, viremia, and the lung lesions than that of the negative control group for 19 days post-challenge (dpc). Additionally, the recombinant virus induced 4.50 ± 3.00 (log_2_) and 8.25 ± 0.96 (log_2_) of neutralizing antibody before and after challenge, respectively. Taken together, this study confirmed that N44S substitution can create an infectious PRRSV that strongly induces neutralizing antibodies. In addition, the vCSL1-GP5-N44S mutant that we produced was confirmed to have potential as a vaccine candidate, showing good safety and protective effects in pigs.

## 1. Introduction

Porcine reproductive and respiratory syndrome (PRRS) was first discovered in the United States in 1987. It was named the “mystery pig disease” and was later discovered in Europe in 1990. The causative agent, porcine reproductive and respiratory syndrome virus (PRRSV), was first isolated in the Netherlands in 1991 and designated Lelystad; a genetically different virus, VR-2332, was isolated in the United States in 1992 [1,2].

PRRSV causes reproductive failure, including stillbirth and autolyzed and mummified fetus in sows, as well as respiratory disease leading to fever, severe dyspnea, anorexia, and lethargy in growing pigs. It also causes additional secondary infections due to immune suppression [3,4]. Various vaccines, including modified live virus (MLV) and killed virus (KV), are commercially available and regarded as a practical way to control PRRS [5,6]. The KV vaccine has advantages from a safety perspective, but it has shown limited efficacy in preventing or reducing symptoms of the disease in tests using young and sow models [7]. Another study that tested KV with hypoglycosylation showed that the candidate could improve the performance of the farm by inducing high levels of neutralizing antibodies [8]. However, MLV has shown protective efficacy against the homologous strain of PRRSV under experimental conditions [7]. In addition, MLV has shown partial protection against heterogeneous strains within the same genotype [6]. However, its efficacy is still not optimal for eradication of the disease in farm environments, and there have been cases of large-scale outbreaks of PRRS in well vaccinated farms using MLV [9,10]. In addition, MLV carries the risk of inducing vaccine-like virulent variants [3,6,11,12].

PRRSV belongs to the same family (Arteriviridae) as lactate dehydrogenase-elevating virus (LDV), equine arteritis virus (EAV), and simian hemorrhagic fever virus (SHFV) [13]. PRRSV is a virus with a single-stranded positive-sense RNA genome, classified into two genotypes: PRRSV-1 (European virus) and PRRSV-2 (North American virus). The two groups showed approximately 50–60% sequence homology [14]. Even within the same genotype, cross-immunity against heterogenous strains is limited in relation to genetic diversity [15].

The genome length is 15.1–15.5 kb, expressed through subgenomic mRNA transcripts of 10 open reading frames (ORFs). ORFs 1a and 1b encode nonstructural proteins for viral replication; ORFs 2–7 encode structural proteins, including glycoprotein (GP) 2a, E, GP3, GP4, GP5, M, N, and GP5a [16]. GP5 and M proteins form hetero-dimeric structures in the envelope and play an important role in infectivity by interacting with the host receptors [17].

The major envelope protein, GP5, is composed of transmembrane regions and a N-terminal ectodomain with several neutralizing antibody epitopes against PRRSV [18,19]. GP5’s ectodomain plays a role in neutralization, comprising 2–5 potential N-linked glycosylation sites [20,21]. Therefore, the glycan around the epitope reduces the PRRSV’s immunogenicity [22,23]. The level of neutralizing antibodies formed after vaccination or infection with field strains is limited because of glycosylation. In addition, protection against subsequent infections is challenging [20].

After vaccination, protection against the PRRSV and induction of a sterile immune system were associated with neutralizing antibody levels [20]. To develop vaccine candidates that induce high levels of neutralizing antibodies, various previous studies have attempted to substitute the potential N-linked glycosylation sites of the PRRSV-2 GP5 protein with alanine (A), aspartic acid (D), lysine (K), or serine (S) [8,20,23,24]. Applying single or multiple substitutions, some cases resulted in obtaining intact PRRS mutants and inducing high levels of neutralizing antibodies in animal trials [8,20,23,24]. However, in other cases, replicable mutant virus could not be obtained [20]. Therefore, detailed research is needed for each virus strain to determine which amino acid substitution at which asparagine (N) residue is appropriate [8,23,25,26].

In our previous study, we developed a chimeric PRRSV by replacing the GP5 ectodomain of PRRS-2 lineage-5 with that of the PRRSV-2 lineage-1 virus using reverse genetic techniques [8]. Using this method, we were able to easily create a vaccine candidate with protective efficacy against MARC-145 cell-unadaptive lineage-1 strain, the currently emerging genotype in Korea. In this study, we created a new vaccine candidate, vCSL1-GP5-N44S, by replacing the 44th asparagine amino acid in the GP5 gene of the chimeric virus with serine, to improve the ability of the parental virus to induce neutralizing antibodies. Subsequently, the novel chimeric mutant virus was inoculated into piglets and evaluated for safety. The inoculated pigs were challenged with the field strain of PRRSV to evaluate their protective efficacy against the homologous PRRS-2 lineage-1 virus. Thus, we reviewed its potential as a vaccine candidate for pigs.

## 2. Materials and Methods

### 2.1. Cells, Viruses and Plasmids

MARC-145 cells were maintained in Dulbecco’s Modified Eagle’s Medium (DMEM) with 10% fetal bovine serum (FBS) (Thermo Fisher Scientific, Cleveland, OH, USA) and 1% Antibiotic Antimycotic Solution (100×) (Thermo Fisher Scientific, Cleveland, OH, USA) [27]. Porcine alveolar macrophages (PAMs) recovered from the lungs of three-week-old piglets from a PRRSV-negative farm [28] maintained in Roswell Park Memorial Institute (RPMI) 1640 medium with same supplements.

In a previous study, we constructed a DNA-launched infectious clone for the PRRS virus isolate CSNA11 (GenBank accession No.: OM777142) and replaced the ectodomain of the ORF5 gene with that of KU-PRRSV-2020-002 (GenBank accession No.: OM037453). KU-PRRSV-2020-002, which is a field isolate, could not induce cytopathic effects (CPE) in cell culture because it was unadaptive to MARC-145 cells. The resulting product was named pCSL1-GP5-wt [8]. CSNA11 and KU-PRRSV-2020-002 are PRRS-2 viruses belonging to lineage-5 and-1, respectively.

### 2.2. Generation of the Tentative Hypoglycosylated Mutant Virus

We constructed a plasmid (pCSL1-GP5-N44S) in which the 44th amino acid, asparagine, in the GP5 protein of pCSL1-GP5-wt was changed to serine using the QuikChange II XL site-directed mutagenesis kit (Agilent, Palo Alto, CA, USA). The primers were designed using the web tool provided by the manufacturer [29].

For transfection, 50 µL of a solution of 1 µg/µL mutant plasmid, Opti-MEM (Thermo Fisher Scientific, Cleveland, OH, USA) medium, and ViaFect™ Transfection Reagent were prepared and incubated at room temperature for 10 min. This solution was inoculated into MARC-145 cells cultured for 24 h at a concentration of 1.5 × 10^5^ cells/mL in a 24-well plate. After incubating for 4 h in a cell incubator at 37 °C and 5% carbon dioxide (CO_2_), 500 µL of DMEM supplemented with 10% FBS was added. The supernatant was collected after 96 h and passed through freshly cultured MARC-145 cells.

### 2.3. Indirect Immunofluorescence Assay (IFA)

Each PRRS virus was diluted to a 0.01 multiplicity of infection (MOI) and inoculated into MARC-145 cells or PAMs. After 2 d, the supernatant was discarded and washed with phosphate-buffered saline (PBS). A solution of acetone and methanol in a 1:1 ratio was dispensed and fixed at room temperature for 10 min. Afterward, the fixative was removed, and the remaining fixative was removed by standing at room temperature for 20 min [30]. A solution of the primary antibody, Mouse anti-PRRSV-2 N protein MAb clone (Median Diagnostics, Chuncheon, Korea), diluted 1:200 in PBS, was dispensed and left at 37 °C for 30 min, followed by washing three times with PBS. A solution of a secondary antibody, anti-mouse IgG Alexa Fluor 488 (Thermo Fisher Scientific, Cleveland, OH, USA), diluted 1:500 in PBS, was dispensed, left at 37 °C for 30 min, washed three times with PBS, and observed under a fluorescence microscope.

### 2.4. Growth Kinetics

To compare the growth kinetics of vCSL1-GP5-wt (a chimeric virus) and vCSL1-GP5-N44S (a hypoglycosylated virus), each stock was diluted with DMEM with 2% FBS to 0.1 MOI. The mixtures were then inoculated into MARC-145 cells, cultured for 24 h, and incubated at 37 °C for 1 h. After washing twice with the medium, a new medium was added; the plate was incubated at 37 °C.

Each well‘s supernatant was recovered at 12, 24, 48, 72, and 96 h after inoculation. The recovered virus was inoculated into newly cultured MARC-145 cells in a 96-well plate using a 10-fold serial dilution method, and the cytopathic effect (CPE) was observed for 6 d. The viral titer was calculated using the Spearman and Kärber method [31].

### 2.5. Animal Experiments

Eight three-week-old castrated piglets were purchased from a PRRSV-free crossbred (large white-landrace-duroc triple cross) herd. After confirming that the pigs were PRRSV negative through serological diagnosis, they were divided into two groups using a random number generator (SPSS Inc., Chicago, IL, USA) and blinded to inspectors. The group size (four piglets per group) was determined based on the study design of a previous study [32]. Each group was relocated to a separate pig isolator inside the facility and allowed to acclimatize for 7 d. Afterward, group 1 was inoculated with vCSL1-GP5-N44S (10^5.5^ TCID_50_/1 mL/dose), while group 2 was inoculated with 1 mL of PBS. At 42 d post-inoculation (dpi), both groups were challenged with KU-PRRSV-2020-002 (10^5.0^ TCID_50_/1 mL/dose), the PRRSV-2 lineage-1 field strain. All inocula were injected into the dorsal part of the neck muscles behind the ear using a 23 G, 1” long syringe.

At 0, 7, 14, 21, 28, 35, and 42 dpi and 7, 14, and 19 d post-challenge (dpc), rectal body temperature and clinical symptoms were measured, and blood was then collected [7,33]. Clinical scoring was measured according to the following criteria: respiratory score: 0 = normal, 1 = mild dyspnea/tachypnea when pressured, 2 = mild dyspnea/tachypnea when relaxed, 3 = moderate dyspnea/tachypnea when pressured, 4 = moderate dyspnea/tachypnea when relaxed, 5 = severe dyspnea/tachypnea when pressured, 6 = severe dyspnea/tachypnea when relaxed; activity score: 0 = normal, 1 = unwilling to move/exaggerated response, 2 = reduced mobility, decreased attention, indolent, or offensive, 3 = recumbency, abnormal rectal temperature; depression score: 0 = alert and interested in feed, 1 = slow to feed or move, 2 = hesitant to feed or move, 3 = anorexia, movement only when stimulated. Serum was separated from the blood samples, and enzyme-linked immunosorbent assay (ELISA), quantitative real-time polymerase chain reaction (RT-qPCR), and serum neutralization tests were performed. Body weight was measured at 0, 14, and 42 dpi and 19 dpc.

The animals were euthanized at 19 dpc; bronchial lymph nodes and lung tissues were collected and subjected to RT-qPCR analysis and scoring for macroscopic and microscopic lung lesions.

There were no inclusion or exclusion criteria, and no animals, experimental units, or data points were excluded from the study. The Institutional Animal Care and Use Committee of Konkuk University approved the animal experiments (No. KU21191).

### 2.6. RT-PCR and Sequencing Analysis

RT-PCR and sequencing were performed to verify nucleotide sequences of the ORF5 region which is encoding GP5 protein. Viral RNA from each sample was extracted using the RNesay mini kit (Qiagen, Hilden, Germany). RT-PCR was then performed. Two primers (ORF5_F 5′-GTCCAACATGTCAAGGAGTT-3′ and ORF5_R 5′-TATATCATCACTGGCGTGTAGG-3′) were used to amplify the ORF5 region. The SuperScript™ III One-Step RT-PCR System with Platinum™ Taq DNA Polymerase (Thermo Fisher Scientific, Cleveland, OH, USA) was used as the reaction solution. cDNA synthesis was carried out at 55 °C for 30 min, cDNA denaturation at 94 °C for 2 min, and gene amplification was repeated 40 times at 94 °C for 15 s, 55 °C for 30 s, and 68 °C for 1 min; the final extension was set at 68 °C for 5 min. PCR was performed using a ProFlex PCR System (Thermo Fisher Scientific, Cleveland, OH, USA). The amplification product was electrophoresed on a 1% agarose gel at 100 V for 30 min and then sequenced (Macrogen, Seoul, Korea). Sequence analysis was performed using Geneious Prime 2022.0.1 (https://www.geneious.com (accessed on 16 March 2022)).

### 2.7. RNA Extraction and RT-qPCR

RNA of each sample was extracted by the method mentioned above; the viral load in individual samples was measured using RT-qPCR. The PRRSV-specific primers and probes were modified from those used in a previous study [34,35,36]. The RT-qPCR reaction mix was prepared according to the manufacturer‘s protocol using the RNA UltraSense One-Step Quantitative RT-PCR System (Thermo Fisher Scientific, Cleveland, OH, USA). For the reaction, cDNA synthesis was carried out at 50 °C for 15 min; cDNA denaturation at 95 °C for 2 min; gene amplification was repeated 45 times at 95 °C for 15 s, 56 °C for 20 s, and 72 °C for 30 s. A LightCycler 96 (Roche, Basel, Switzerland) was used for the tests. The limit of quantitation (LOQ) of this test was calculated as 1.2 log (TCID_50_/mL) from 10 repeated measurements, which is the level at which the virus could not be isolated.

### 2.8. ELISA

The IDEXX PRRS X3 Ab Test (IDEXX Laboratories, Inc., Columbus, OH, USA) was used to measure the concentration of PRRSV-specific IgG in the serum. The experiment was conducted according to the manufacturer’s guidelines. Following the reaction, optical density (OD) at 650nm was acquired using a microplate reader (TECAN, Männedorf, Switzerland) and the sample-to-positive (S/P) ratio was determined using the OD value. A positive result was defined as an S/P ratio of ≥0.4.

### 2.9. Pathological Analysis

Lung tissue was collected from pigs 19 dpc. For macroscopic lung lesions, the anterior, middle, caudal, and accessory lobes of the lung were visually observed on both sides, and the scores were measured according to the progression of the lung lesion [37]. Each lobe of the lung was collected and fixed with 10% formalin solution (Sigma-Aldrich, Hamburg, NY, USA) to observe microscopic lesions. Tissue sections were prepared and stained with hematoxylin and eosin (H&E; MIBiotec, Seoul, Korea). The tissue slides were randomly numbered, and the scores were measured by repeated observation by three observers under a 200× magnification microscope.

### 2.10. Serum Neutralization Test

A serum neutralization test was performed as previously described [38]. MARC-145 cells were seeded in each well of a 96-well plate at a concentration of 3 × 10^5^/mL and used after culturing for 24 h. Serum samples were diluted 2–64 times by two-fold serial dilution in DMEM supplemented with 2% fetal bovine serum. Approximately 60 μL of each diluted serum and 60 μL of target PRRSV (vCSL1-GP5-N44S) diluted to a concentration of 2.5 × 10^3^ TCID_50_/mL were mixed and incubated at 37 °C for 1 h. After removing the medium from the prepared 96-well plate, 100 μL of the serum–virus mixture was inoculated into each well and incubated for 1 h. The inoculum was removed, and 100 μL of fresh medium was added. The cytopathic effect was individually observed for 6 d to evaluate the neutralizing antibody level. The titers were determined as the reciprocal of the dilution that produced a 50% or higher reduction in the wells. The neutralization test was performed in three replicates for each serum sample [37].

### 2.11. Statistical Analysis

All statistical analyses were performed using the non-parametric Mann–Whitney test and calculated using IBM SPSS Statistics 25 (IBM Corp., Armonk, NY, USA). A *p*-value difference ≤ 0.05 was considered statistically significant. All graphs were prepared using GraphPad Prism software (GraphPad Software, Inc., San Diego, CA, USA).

## 3. Results

### 3.1. Tentative Hypoglycosylated Mutations and Virus Rescue

pCSL1-GP5-N44S was constructed by substituting serine with the 44th amino acid asparagine (N) in the GP5 protein of pCSL1-GP5-wt (Figure 1A). After transfection with pCSL1-GP5-N44S, the transfection reaction supernatant was designated as passage 0 and successively passaged to the newly prepared cells. CPE was observed after the 2nd passage. Finally, the new mutant virus replication was visually confirmed using indirect immunofluorescence antibody analysis after infecting confluent MARC-145 cells with 0.01 MOI of the virus for two days (Figure 1B,C). The rescued tentative hypoglycosylated virus was named vCSL1-GP5-N44S. When each passage was examined by RT-qPCR and sequencing techniques, the introduced mutation remained stable until the seventh successive passage (Appendix A).

### 3.2. Characteristics of Tentative Hypoglycosylated Viruses

To compare the growth potential of vCSL1-GP5-wt and vCSL1-GP5-N44S, each virus was inoculated with MARC-145 cells and recovered at 0, 12, 24, 48, 72, and 96 h after inoculation. The recovered samples were then titrated. The highest titers of all viruses were observed at 72 h after inoculation; vCSL1-GP5-N44S reached 5.82 ± 0.09 logTCID_50_/mL and vCSL1-GP5-wt reached 6.26 ± 0.18 logTCID_50_/mL. The vCSL1-GP5-N44S titers were lower than those of and vCSL1-GP5-wt at all time points (Figure 1D). The viruses showed no statistically significant differences at each time point.

In addition, the virus was inoculated into PAMs, and the degree of replication was compared in MARC-145 cells. Regarding the titer measurement at 72 h after inoculation, the virus formed a titer 2884 times (vCSL1-GP5-wt, 2.80 ± 0.14 logTCID_50_/mL), 155 times (vCSL1-GP5-N44S, 3.63 ± 0.08 logTCID_50_/mL) lower in PAM than in MARC-145 cells (Figure 1E). The titers in PAMs and MARC-145 cells of both viruses showed statistically significant differences (*p* < 0.05).

### 3.3. Pathogenicity and Safety Assessment of Tentative Hypoglycosylated Viruses

To evaluate the safety of the hypoglycosylated virus, viremia, rectal body temperature, clinical signs, and weight gain were assessed for 42 dpi. Viremia was measured using RT-qPCR with the RNA extracted from serum obtained through blood collection. The virus was not detected in any of the groups during the observation period (Appendix A). Clinical symptoms were not observed in the tentative hypoglycosylated virus inoculation or the negative control groups (Appendix A).

Rectal body temperature measurements of groups 1 and 2 showed 38.8 ± 0.5 °C and 38.7 ± 0.4 °C, respectively. The temperature of the group inoculated with the tentative hypoglycosylated virus was relatively higher than that of the negative control group at each time point, except at 21 d; however, they were within the normal range. There was no statistically significant difference between the two groups (*p* > 0.05, Figure 2A).

The average daily weight gain was obtained by measuring the body weight before the test and at 42 d; the mean average daily weight gain values of groups 1 and 2 were 0.41 ± 0.09 and 0.42 ± 0.11 kg/day, respectively. There were no statistically significant differences in the values between the groups (*p* > 0.05, Figure 2B).

### 3.4. Evaluation of the Protective Efficacy of Tentative Hypoglycosylated Viruses

To evaluate the protective effect of the hypoglycosylated virus, both groups were challenged with field PRRSV-2 lineage-1 (PRRSV KU-PRRSV-2020-002) isolates at 42 dpi and observed for 19 d. During the observation period, the body temperature of the mutant inoculated group (39.2 ± 0.6 °C) was lower than that of the negative control group (39.6 ± 0.6 °C, Figure 3A) through all time points. At 3 dpc, the body temperature of group 1 (38.9 ± 0.2 °C) was significantly lower than that of group 2 (39.5 ± 0.4 °C, *p* ≤ 0.05). All pigs in group 1 had body temperatures within the normal range, but all pigs in group 2 had temperatures over 40 °C at 14 dpc. Viremia was measured using RT-qPCR, and the viremia level in the inoculated group was lower than that in the negative group (Figure 3B). Seven days post-challenge, the viremia of the inoculation group (0 ± 0 logTCID_50_/mL, *p* ≤ 0.05) was significantly lower than that of the control group (1.32 ± 0.45 logTCID_50_/mL). Comparing the area under the curve (AUC) of the graph showing viremia, the area of group 1 (3.03 ± 2.37) was significantly lower than that of group 2 (12.62 ± 4.26, *p* ≤ 0.05).

After euthanizing all the piglets on 19 dpc, macroscopic and microscopic lung lesions were scored. In the comparison of macroscopic scores, the scores for group 1 (10.13 ± 6.11) were statistically lower than those of group 2 (18.41 ± 3.36, *p* ≤ 0.05, Figure 3C). The microscopic lesion scores were also significantly lower in the tentative hypoglycosylated virus inoculation group (1.30 ± 0.32) than in the control group (2.18 ± 0.75, *p* ≤ 0.05, Figure 3D).

### 3.5. Neutralizing Antibody Measurement

We tested only the ELISA-positive samples for serum neutralization to measure the neutralizing antibodies from the tentative hypoglycosylated virus. In the case of neutralizing antibodies against vCSL1-GP5-N44S (Figure 4A), group 1 piglets showed seroconversion at 28 dpi, and the neutralizing antibody titer increased to 4.50 ± 3.00 (log_2_) at 42 dpi. After the challenge, the neutralizing antibody level in the inoculated group peaked at 8.25 ± 0.96 (log_2_) at 7 and 19 dpc. However, the value in group 2 was not detected until the last day of the safety experiment and was 1.75 ± 2.06 (log_2_) at 19 dpc. The antibody titers showed a significant difference between the two groups post-challenge (*p* ≤ 0.05).

Regarding the neutralizing antibodies against vCSL1-GP5-wt (Figure 4B), group 1 piglets showed seroconversion at 35 dpi, and the neutralizing antibody titer reached 1.00 ± 2.00 (log_2_) at 42 dpi. After the challenge, the neutralizing antibody level in the inoculated group peaked at 4.00 ± 2.83 (log_2_) at 19 dpc. However, the value in group 2 was 0.75 ± 1.50 (log_2_) at 14 dpc and 1.25 ± 2.50 (log_2_) at 19 dpc. The antibody titers did not show any significant difference between the two groups post-challenge.

## 4. Discussion

In this study, we successfully constructed a tentative hypoglycosylated vaccine candidate by substituting the 44th asparagine (N) of the GP5 gene with serine by genetically engineering the PRRS-2 lineage-1 and lineage-5 chimeric viruses obtained from a previous study [8]. In addition, we clearly presented the potential as a vaccine candidate through safety and protective efficacy tests in vivo. These results confirm that modifying the 44th glycosylation site can increase the induction of high levels of neutralizing antibodies. In particular, we propose that replacing the 44th asparagine with serine not only increases efficacy but also does not interfere with the virus’s kinetics.

Although studies have shown a protective ability post viral challenge, even if neutralizing antibodies are not present after vaccination, sizable SVN can protect against PRRSV infection [15,39,40,41]. In a study in which neutralizing antibodies were transferred and then challenged using a young piglet model, it was suggested that neutralizing antibodies in blood at 3 (log_2_) or higher could clear viremia [42]. Other studies found that high levels of neutralizing antibodies were associated with a low viral load in the blood post-challenge [25,43]. In this study, neutralizing antibodies to homologous strain in pigs inoculated with vCSL1-GP5-N44S reached a maximum of 4.50 ± 3.0 (log_2_) at 42 dpi and reached 8.25 ± 0.96 (log_2_) post-challenge. In addition, the inoculation group cleared viremia faster than the control group, and the viremia AUC was also significantly different. Therefore, in accordance with the results of previous studies, it was confirmed that inducing high levels of neutralizing antibodies after vaccination guarantees protection against infection. On the other hand, the neutralizing antibody titers against vCSL1-GP5-wt were measured as low, at 1.00 ± 2.00 (log_2_) at 42 dpi and 4.00 ± 2.83 (log_2_) at 19 dpc, compared to those for the homologous virus. This phenomenon is consistent with other studies on glycosylation alteration of PRRSV GP5, and is still noticeably meaningful, inducing an increase of at least 3 (log_2_) post-challenge [20,42,43,44].

The slow and weak formation of neutralizing antibodies against the PRRSV is related to the N-linked glycosylation site surrounding the major epitope (37–44th amino acid) in the ectodomain of GP5 [20,21]. Previous studies have focused on modifying glycosylation by introducing mutations into GP5. One study showed that applying single or multiple substitutions of N34 and N51 glycosylation sites in the PRRS-2 GP5 gene with alanine (A), the ability to induce neutralizing antibodies and the sensitivity to neutralizing antibodies increased [20]. In a similar study, as a result of various applications of S32N, N35S, N44K (lysine), and N41S substitutions to the gene, the ability to induce neutralizing antibodies was improved compared to the original virus [24]. Similar to our study, when the GP5 gene of the chimeric PRRS vaccine candidate was applied to N32A and N51A mutations, the ability to induce neutralizing antibodies was improved compared to the original virus [23]. On the other hand, when N44 was substituted with A, the virus was not rescued in vitro, whereas when substituted with K, an intact virus was obtained, and neutralizing antibodies were induced to a higher level than with the parent virus in vivo [20,24]. In this study, we observed that a very high level of neutralizing antibody was induced in vivo without harming the infectivity of the virus. In this context, the N44S substitution introduced by us can tentatively be seen as hypoglycosylating the GP5 gene. Therefore, it has been demonstrated that modifying N-linked glycosylation is a useful tool for developing vaccine candidates and substituting the 44th N-linked glycosylation site with serine is preferable among the various N-linked glycosylation sites.

Immunity to PRRSV is better established when using a vaccine that replicates in vivo [7,39]. However, the level of viremia after vaccination does not appear to be proportional to the formation of immunity. In a study, a synthetic PRRS vaccine candidate showed high levels (8–9 log genome copies/mL) of viremia, similar to the pathogenic wild-type virus, subsequently showing protection in a challenge test [39]. In other studies, vaccine candidate strains showed protective ability in a challenge infection after a lower level of viremia than that of the highly pathogenic control virus [25,40,43,45]. Meanwhile, in another study, half of the vaccinated pigs were induced with viremia, while the others were not, and neutralizing antibodies appeared in all animals [23]. In this study, pigs inoculated with vCSL1-GP5-N44S did not show viremia, but neutralizing antibodies were induced and were protective post-challenge. Therefore, it can be said that sufficiently attenuated viruses can induce neutralizing antibodies regardless of the level of viremia. This relationship can also be observed in other viruses, such as Vesicular Stomatitis New Jersey Virus [46]. The most reasonable hypothesis was that immunity was formed because the virus replicated in the lymph organs (e.g., the tonsils), while the virus did not replicate sufficiently in the PAMs to establish systemic infection [23]. This hypothesis is supported by the fact that the titer of the viruses used in this study was reduced when they propagated in PAM cells.

Although the components of the immune system responsible for protective immunity against PRRSV is not fully understood yet, the role of PRRS vaccine for cell-mediated immunity appears to be as important as that for humoral immunity [47,48]. For example, several studies on vaccine candidates have shown protective efficacy post-challenge, even if neutralizing antibodies were not observed after vaccination [39,40,49]. The T cell response to PRRSV is mainly measured by the level of PRRSV-specific INFγ-secreting cells, which is measured primarily using the enzyme-linked immunospot (ELISPOT) assay [47,48]. One study suggested that the protective efficacy of vaccines correlates with the ability to induce PRRS-specific IFNγ-secreting cells(SC) [50]. In another study, the PRRS-2 lineage-5 vaccine candidate induced IFNγ-SC in high proportion and showed protective ability in a lineage-1 heterologous challenge [45]. Natural killer (NK) cells, which trigger early IFNγ responses to PRRSV infection, down-regulate CD163 expression in macrophages and inhibit IL-10 production, thereby reducing susceptibility to PRRSV infection [48]. Therefore, the ability to induce cell-mediated immunity is one of the key factors in evaluating the potential of vaccine candidates. In this study, we did not measure the cell-mediated immunity-related values of vCSL1-GP5-N44S, but further study is required.

Developing a vaccine against PRRSV is challenged by the genetic variability associated with viral replication properties [15]. PRRS-2 is divided into nine lineages based on the genome, and the defense against different lineages is partial, depending on the vaccine candidate strain [43]. Even with two vaccine candidates of the same lineage, one study showed different protective abilities against the same challenge strain [45]. Therefore, to verify the potential of vaccine candidates, their protective ability against heterologous challenges should be confirmed. The backbone virus used in this study, vCSL1-GP5-wt, was prepared by replacing the ectodomain of vCSL1, belonging to lineage-5, with the lineage-1 strain [8]. A report on chimeric vaccine candidates made by different lineage viruses, similar to our candidate, suggested a defense against a heterogenous challenge [43,45]. We expect that vCSL1-GP5-N44S can defend against lineage-1 and -5 simultaneously, but further testing is needed [8].

A major risk of using live vaccines against PRRSV is the emergence of pathogenic revertants and recombinants due to the live vaccines circulating within the farm [11,23,51]. Therefore, field tests on the degree of shedding, the degree of spreading to the comingle group, and the recovery of pathogenicity over a long period are required for vaccine candidates. As a result of testing a vaccine virus with two amino acid substitutions in GP5 on a commercial farm, one of them reverted to the same as the wild-type virus [23]. Shedding is induced because of the systemic distribution through macrophages, with the appearance of viremia after PRRSV infection [52,53]. In this study, no viremia was observed in pigs inoculated with vCSL1-GP5-N44S. The chimeric GP5 mutants prepared in a similar way as in our study also showed low levels of viremia, and shedding and spreading were not observed [23,25]. We also expect that vCSL1-GP5-N44S is safe against shedding and circulation in field conditions, but further testing is required.

## 5. Conclusions

vCSL1-GP5-N44S is a chimeric mutant virus with an enhanced ability to induce neutralizing antibodies through tentative hypoglycosylation and is a vaccine candidate for PRRSV-2 lineage-1. In the safety trial, clinical symptoms and body temperature changes were not significantly different from those of the negative control group. The inoculated group showed significantly higher neutralizing antibody levels, lower viremia AUC levels, and lower lung lesion scores compared with the control group post-challenge. Therefore, we propose that modifying the N-linked glycosylation site of the GP5 gene is a useful tool for creating vaccine candidates that induce high levels of neutralizing antibodies. Thus, we present, for the first time, that by replacing the 44th asparagine with serine, we can create a hypoglycosylated vaccine candidate without damaging the infectivity of PRRSV. Additionally, vCSL1-GP5-N44S showed good performance concerning safety and efficacy, confirming its potential as a vaccine candidate for the PRRSV-2 strain-1.

## Figures and Tables

**Figure 1 vetsci-10-00191-f001:**
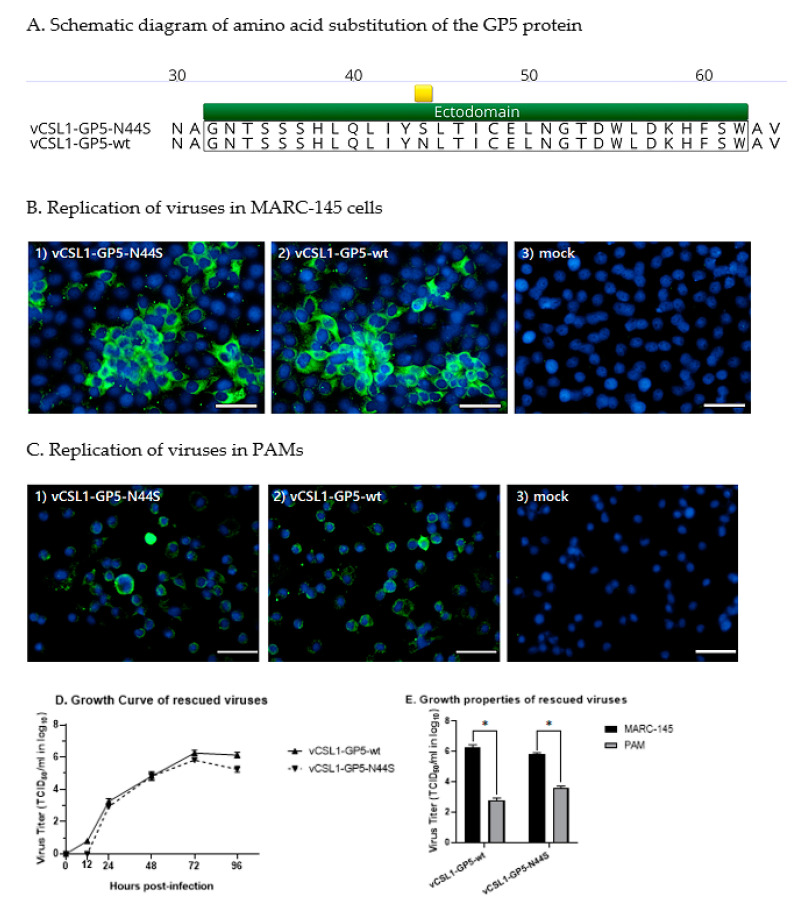
Introduction of mutations in GP5 protein and comparison of replication characteristics with backbone viruses. (**A**) The amino acid sequence alignment of GP5 protein between the backbone virus, vCSL1-GP5-wt, and the novel virus, vCSL1-GP5-N44S. The 44th substitution site is indicated by a yellow band. (**B**,**C**) Replication of mutant virus detected using indirect immunofluorescence assay in MARC-145 cells or PAM cells. The green signal is PRRSV, the blue signal is the nucleus, and the white band represents the 50 µm scale. (**D**) Comparison of the one-step growth curves of the backbone and novel viruses in MARC-145 cells. (**E**) Comparison of maximal titers of backbone and novel viruses in MARC-145 and PAM cells 72 h after inoculation. The symbol “*” indicates statistically significant differences (* *p* ≤ 0.05).

**Figure 2 vetsci-10-00191-f002:**
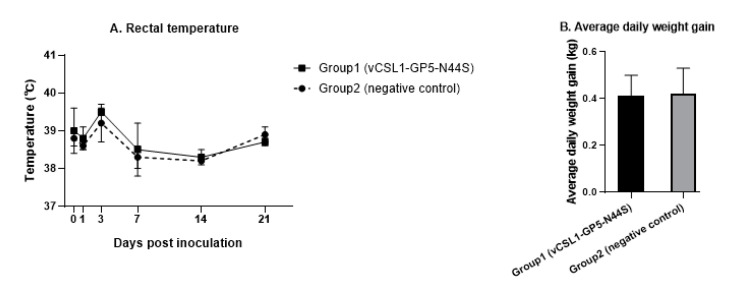
Results of the safety evaluation of vCSL1-GP5-N44S. Four-week-old pigs were assigned to two groups, and each group was intramuscularly injected with 1 mL of vCSL1-GP5-N44S at the dose of 10^5.5^ TCID_50_/mL (group 1) and 1 mL of PBS (group 2). (**A**) Rectal temperature was measured 0–21 d post-inoculation. (**B**) Average daily weight gain was calculated by measuring body weight at 0 and 42 d post-inoculation.

**Figure 3 vetsci-10-00191-f003:**
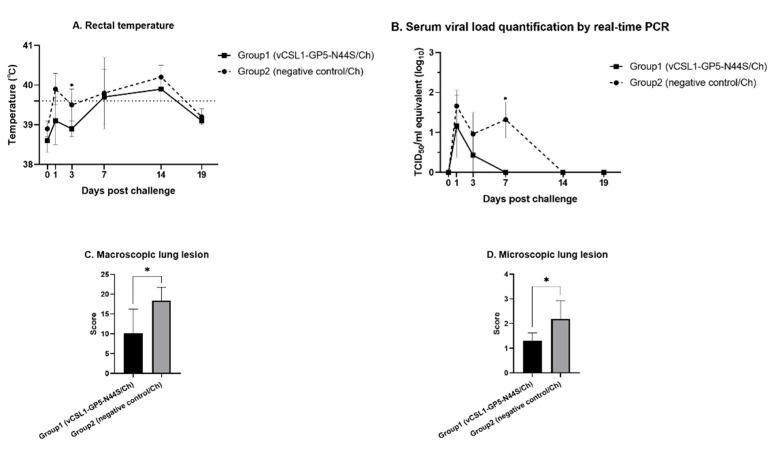
Results of the efficacy evaluation of vCSL1-GP5-N44S. All groups used for safety tests were challenged by intramuscular route inoculation with 10^5.0^ TCID_50_/1 mL/dose of KU-PRRSV-2020-002 at 42 d post-inoculation. (**A**) Rectal temperature measured from 0 to 19 d post-challenge. (**B**) Viral RNA load in serum collected from 0 to 19 d post-challenge, determined by PRRS-specific RT-qPCR. (**C**) Macroscopic lung lesion and (**D**) microscopic lung lesion scores were evaluated at necropsy. The symbol “*” indicates statistically significant differences from the negative control (* *p* ≤ 0.05).

**Figure 4 vetsci-10-00191-f004:**
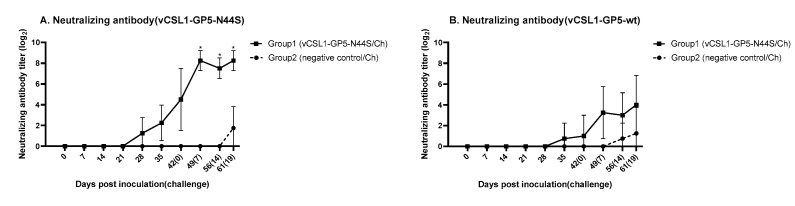
Changes in serum neutralizing antibodies after inoculation or challenge. Serum obtained 0, 7, 14, 21, 28, 35, and 42 d post-inoculation and 7, 14, and 19 d post-challenge was measured by a serum neutralization test against vCSL1-GP5-N44S (**A**) or vCSL1-GP5-wt (**B**). The symbol “*” indicates statistically significant differences from the negative control (* *p* ≤ 0.05).

## Data Availability

Data are available from the corresponding author upon reasonable request.

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
