# Peer review of "Porcine Reproductive and Respiratory Syndrome Virus Engineered by Serine Substitution on the 44th Amino Acid of GP5 Resulted in a Potential Vaccine Candidate with the Ability to Produce High Levels of Neutralizing Antibody"

_vetsci, 2023, doi:10.3390/vetsci10030191_

Round 1
Reviewer 1 Report
The experiments are well designed and the article is well written, carefully edited.
Here are some minor comments on some particular points to clarify:
- Lines 21-27: Simple Summary is missing, write it, or delete
- line 47: clarify what was discovered in the US in 1987. Not the virus, but the disease. In line 50 you write, that the virus was isolated only in 1992.
- line 89: It is confusing to mention a hypoglycosylated mutant virus. Please clarify whether it has any relation to the vCSL1-GP5-N44S strain, or it is only an other earlier example. Mentioning the strain ID or the lab can be helpful.
- lines 107 and 111 - almost the same, try to simplify . Moreover: 'Antibiotic-Antimycotic' is the product name written like this. This product and FBS are both 'Gibco' products which is now a brand name of Thermo Fisher, so use one of them, but uniformly
- lines 187-189: However it is mentioned in lines 75-76, please mention here as well, like 'ORF5 region (encoding GP5)'
- line 263: Add: 'There were no statistically significant differences ........'
- Table 1: This is important data, but NOT waste the space for two tables full with zero values! Delete the tables. If it is necessary, expand the statement in line 283.
Table 2 is not referred in text in point 3.3. Similarly to Table 1: write down the negative result and delete Table 2!
After these small corrections the article will be more understandable and ready for publication.
Author Response
We would like to express our sincere appreciation for your meticulous review of our manuscript and valuable feedback provided. Your comments have greatly contributed to the improvement of the quality of our work. Thank you for your time and effort in reviewing our paper.
Here is our response to your comment:
Q1: Lines 21-27: Simple Summary is missing, write it, or delete
A1: We omitted the Simple Summary in the process of writing the manuscript. We have added it in response to your suggestion. Thank you for your comment. (Line 21-30)
Q2: line 47: clarify what was discovered in the US in 1987. Not the virus, but the disease. In line 50 you write, that the virus was isolated only in 1992.
A2: In the process of writing the manuscript, we confusedly wrote the name of the disease and the phrase referring to the causative agent. The manuscript has been updated to make the meaning clearer. We appreciate your comments. (Line 49-53)
Q3: line 89: It is confusing to mention a hypoglycosylated mutant virus. Please clarify whether it has any relation to the vCSL1-GP5-N44S strain, or it is only an other earlier example. Mentioning the strain ID or the lab can be helpful.
A3: The sentence we wrote was not as clear as you suggested. We have revised the manuscript. Thank you for your feedback. (Line 89-98)
Q4: lines 107 and 111 - almost the same, try to simplify . Moreover: 'Antibiotic-Antimycotic' is the product name written like this. This product and FBS are both 'Gibco' products which is now a brand name of Thermo Fisher, so use one of them, but uniformly
A4: We have edited the paragraphs to be concise and corrected old expressions like Gibco. We appreciate your comments. (Line 112-116)
Q5: lines 187-189: However it is mentioned in lines 75-76, please mention here as well, like 'ORF5 region (encoding GP5)'
A5: We have corrected the wording as per your suggestion. This made the meaning clearer. thanks for your advice (Line 190-121)
Q6: line 263: Add: 'There were no statistically significant differences ........'
A6: In fact, there was a statistical difference, and the missing part was additionally described. The same method as described in "2.11 Statistical Analysis" was used. Figure 1E was also modified. Thank you for your thoughtful advice. (Line 272-273)
Q7: Table 1: This is important data, but NOT waste the space for two tables full with zero values! Delete the tables. If it is necessary, expand the statement in line 283.
Table 2 is not referred in text in point 3.3. Similarly to Table 1: write down the negative result and delete Table 2!
A7: We also felt that the table filled with zeros was a waste of space. However, We had concerns about whether it was appropriate to describe that viremia or clinical signs were not observed without presenting the tables. Therefore, instead of deleting them, I moved them to the supplementary materials (Line 476-481). We are wondering if this is sufficient to apply your advice to the manuscript
We have carefully reviewed and revised our manuscript in response to your comments, and we believe that we have addressed all the concerns you raised. However, we would be grateful if you could kindly check the revised manuscript once again to ensure that everything is in order.
Thank you very much for your time and effort in reviewing our paper. We greatly appreciate your contributions to the improvement of our work.
Reviewer 2 Report
I reviewed the manuscript entitled “Porcine reproductive and respiratory syndrome virus engineered by serine substitution on the 44th amino acid of GP5 protein is not viremic in piglets but induces high levels of neutralizing antibody”. In this study authors successfully developed a potential vaccine candidate of PRRSV with the ability to induce high levels of neutralizing antibodies.
Overall, I think it is an interesting study for the field of PRRSV with applications for the development of vaccine candidates for other viral diseases. These are some of my suggestions to improve the quality of this manuscript.
Title: I would suggest the authors to modify the title of this study for something like: “Porcine reproductive and respiratory syndrome virus engineered by serine substitution on the 44th amino acid of GP5 resulted in a potential vaccine candidate with the ability to produce high levels of neutralizing antibody”. The reason for this suggestion is because I consider that the absence of viremia is not a condition for the lack of production of neutralizing antibodies (see https://doi.org/10.3389/fmicb.2020.01123). There are examples like in VSV where highly attenuated mutants can induce high levels of neutralizing antibodies in absence od viremia. I consider that this fact should be part of the discussion.
Introduction: I consider that this section should be improved. In my opinion, the main subject of this study is not properly explained in this section. There are multiple studies referenced in the discussion section starting from line 371 that may fit better in the introduction section. Please explain better the concept about glycosylation in PRRSV supported by previous studies, highlighting the reasoning behind your methodology strategy.
Why other samples were not considered to evaluate the viral shedding like saliva or oral swabs?
Author Response
We would like to express our sincere appreciation for your meticulous review of our manuscript and valuable feedback provided. Your comments have greatly contributed to the improvement of the quality of our work. Thank you for your time and effort in reviewing our paper.
Here is our response to your comment:
Q1: Title: I would suggest the authors to modify the title of this study for something like: “Porcine reproductive and respiratory syndrome virus engineered by serine substitution on the 44th amino acid of GP5 resulted in a potential vaccine candidate with the ability to produce high levels of neutralizing antibody”. The reason for this suggestion is because I consider that the absence of viremia is not a condition for the lack of production of neutralizing antibodies (see https://doi.org/10.3389/fmicb.2020.01123). There are examples like in VSV where highly attenuated mutants can induce high levels of neutralizing antibodies in absence od viremia. I consider that this fact should be part of the discussion.
A1: We appreciate your suggestion regarding the title. As you pointed out, considering various contexts, we think that the original title of this paper is somewhat inappropriate. Therefore, we have modified the title. We have also included your suggestion regarding the discussion in the manuscript. (Line 407-410)
Q2: Introduction: I consider that this section should be improved. In my opinion, the main subject of this study is not properly explained in this section. There are multiple studies referenced in the discussion section starting from line 371 that may fit better in the introduction section. Please explain better the concept about glycosylation in PRRSV supported by previous studies, highlighting the reasoning behind your methodology strategy.
A2: As per your suggestion, upon reevaluation of our introduction, we realized that our explanation of the rationale for creating a deglycosylation mutant or the strategy for applying substitutions was lacking. Therefore, we have added a brief explanation of the specific method for applying hypoglycosylation using the reference you mentioned (Line 89-98). The detailed comparison with our research findings is included in the Discussion section. We appreciate your advice.
Q3: Why other samples were not considered to evaluate the viral shedding like saliva or oral swabs?
A3: We agree that shedding is important factor in researching vaccine candidates. In the field of PRRS virus, we believe it is appropriate to analyze nasal, oral, and fecal swabs. However, for this study, we prioritized verifying whether vCSL1-GP5-N44S is working as intended. Therefore, we mainly observed viremia, clinical signs, neutralizing antibody levels, and other factors. Through this study, we gained confidence that our vaccine candidate has value for further research. Therefore, we are currently conducting follow-up research, which includes the analysis of shedding. As a result, we would like to address the shedding of vCSL1-GP5-N44S in future publications.
We have carefully reviewed and revised our manuscript in response to your comments, and we believe that we have addressed all the concerns you raised. However, we would be grateful if you could kindly check the revised manuscript once again to ensure that everything is in order.
Thank you very much for your time and effort in reviewing our paper. We greatly appreciate your contributions to the improvement of our work.
Reviewer 3 Report
The work by Choi et al. described the biological properties in vivo of a recombinant virus: vCSL1-GP5-N44S.
Previous work suggested that the N-linked glycans covering GP5 neutralizing epitopes of PRRSV act as a sheath blocking the production of neutralizing antibodies.
In this work, researchers created the recombinant virus vCSL1-GP5-N44S , constructed by subsituting serine (S) with the 44th asparagine (N) on the GP5 ectodomain of PRRSV-2 lineage-1.
To evaluate the recombinant PRRSV, in vivo experiments were performed in piglets. The recombinant virus group showed no viremia until 42 days post-immunization, and no differences with the control grojup were observed in term of RT and average daily weight. On the 42 dpi, both groups were challenged with the wild-type PRRSV. The recombinant PRRSV group showed lower RT, viremia, and the lung lesions than that of the non-immunised control group for 19 days dpc. In addition, vCSL1-GP5-N44S induced higher levels of neutralizing antibody before and after challenge. Overall, this work shoed that vCSL1-GP5-N44S strongly induced neutralising antibodies, and showed good safety and protective effecacy against challenge with wt PRRSV strain in pigs.
Overall this work is devoid of major weeknesses. I found it well written, well structured ans easy to read.
There are few minor point I would suggest before publication:
Line 210-213. Did you read OD (optical density)? How?
Line 247: ‘profileration’ should be substituted with ‘replication’
3.1 Which MOI was used???
Fig 1.D, Figure 1:E. There are statistical differences between PRSSV isolates?
Line 407-419. I suggest you to cyte more recent work about PRRSV cell mediated immunity (these are two more recent review, but there are others… Razzuoli et al., 2022 Pathogens; Rahe et al., 2017 Viruses)
Author Response
We would like to express our sincere appreciation for your meticulous review of our manuscript and valuable feedback provided. Your comments have greatly contributed to the improvement of the quality of our work. Thank you for your time and effort in reviewing our paper.
Here is our response to your comment:
Q1: Line 210-213. Did you read OD (optical density)? How?
A1: According to manufacturer’s manual, we measured optical density (OD) at 650nm and the sample-to-positive (S/P) ratio was determined using the OD value. We thank you for pointing out the omissions. The manuscript has been corrected. (Line 271-220)
Q2: Line 247: ‘profileration’ should be substituted with ‘replication’
A2: We agree with your suggestion that "replication" is a better term than "proliferation". We have incorporated this change into the manuscript. (Line 254)
Q3: 3.1 Which MOI was used???
A3: The confluent MARC-145 cells were infected with 0.01 MOI of the virus for 2days. Thanks for pointing out the missing information. We have edited the manuscript (Line 254-256)
Q4: Fig 1.D, Figure 1:E. There are statistical differences between PRSSV isolates?
A4: (Figure 1D) The viruses showed no statistically significant differences at each time point (Line 266-267). (Figure 1E) The titers in PAMs and MARC-145 cells of both viruses showed statistically significant differences(p<0.05). What you pointed out was also pointed out by other reviewers, and we updated the manuscript. Thank you for your comments. (Line 266, 272).
Q5: Line 407-419. I suggest you to cyte more recent work about PRRSV cell mediated immunity (these are two more recent review, but there are others… Razzuoli et al., 2022 Pathogens; Rahe et al., 2017 Viruses)
A5: We have come to realize that there is a need to improve the paragraph that you pointed out while reading the literature you introduced. We have removed outdated references such as the literature published in 2008 and made some changes to the content. We would appreciate it if you could review the content once again. We are grateful for your suggestion. (Line 415-430)
We have carefully reviewed and revised our manuscript in response to your comments, and we believe that we have addressed all the concerns you raised. However, we would be grateful if you could kindly check the revised manuscript once again to ensure that everything is in order.
Thank you very much for your time and effort in reviewing our paper. We greatly appreciate your contributions to the improvement of our work.
Round 2
Reviewer 2 Report
I like to thank the authors for their responses, at this point, I don't have more concerns about this study.
Author Response
We would like to express my sincere appreciation for your valuable suggestions on my paper. Your insights and feedback have been extremely helpful in improving the quality and clarity of my work.
Thank you once again for your time and effort in reviewing my paper. Your input has been instrumental in enhancing the overall quality of our research.